# Evaluation of Totarol for Promoting Open Wound Healing in Dogs

**DOI:** 10.3390/vetsci11090437

**Published:** 2024-09-16

**Authors:** Somphong Hoisang, Supranee Jitpean, Suvaluk Seesupa, Phanthit Kamlangchai, Tossawarn Makpunpol, Pimsiri Ngowwatana, Saikam Chaimongkol, Duangdaow Khunbutsri, Jeerasak Khlongkhlaeo, Naruepon Kampa

**Affiliations:** 1Veterinary Teaching Hospital, Faculty of Veterinary Medicine, Khon Kaen University, Khon Kaen 40002, Thailand; sompho@kku.ac.th (S.H.); phankam@kku.ac.th (P.K.); saikam_ch@kkumail.com (S.C.); duankh@kku.ac.th (D.K.); jeechl@kku.ac.th (J.K.); 2Division of Surgery, Faculty of Veterinary Medicine, Khon Kaen University, Khon Kaen 40002, Thailand; supraneeji@kku.ac.th (S.J.); suvalukse@kku.ac.th (S.S.); 3Residency Training Program in Veterinary Surgery, Veterinary Teaching Hospital, Faculty of Veterinary Medicine, Khon Kaen University, Khon Kaen 40002, Thailand; tossawarn@kkumail.com (T.M.); pimsiri.n@kkumail.com (P.N.)

**Keywords:** adjunctive treatment, *Podocarpus totara*, totarol, veterinary, wound care

## Abstract

**Simple Summary:**

Standard wound management involving topical antibiotics or essential oils can enhance the healing process. Organic plant oil extracts, such as totarol, have demonstrated efficacy in wound healing as adjunctive treatments. This study focuses on evaluating the antibacterial properties of totarol and its efficacy in clinical wound healing in dogs. This result reveals that totarol exhibited antimicrobial activity against both standard pathogens and clinical wound pathogens. Clinically, the use of totarol as an adjunctive therapy significantly improved wound healing, as indicated by a greater percentage of wound area reduction.

**Abstract:**

This study investigates the susceptibility of common pathogens to totarol and assesses its clinical effectiveness in promoting wound healing in client-owned dogs with open wounds. Twenty-three client-owned dogs with open wounds were divided into two groups: (1) the treatment group (T-group) and (2) the control group (C-group). Clinical samples were collected from the wounds for the bacterial identification and determination of the minimum inhibitory concentrations (MICs) of totarol. In the T-group, wounds were treated with standard wound care together with the application at a dosage of 0.3 mL (two sprays) of commercial totarol product per 25 cm^2^ of the wound area. The C-group received only standard wound care. This in vitro study found that totarol exhibited antimicrobial activity against both standard pathogens and clinical wound pathogens. The MIC values of totarol dissolved in absolute ethyl alcohol were 4 µg/mL for Gram-positive pathogens and ranged from 256 to 512 µg/mL for Gram-negative pathogens. However, the MIC values of the commercial totarol product ranged from 512 to 1024 for both Gram-positive and Gram-negative pathogens. Clinically, the use of a commercial totarol product as an adjunctive therapy significantly improved wound healing, as indicated by a greater percentage of wound area reduction (*p* < 0.05). From day 2 to day 7 of the treatment, the percentage of wound area reduction differed significantly between the T-group and the C-group. At the end of the study, the average percentage of wound area reduction was 69.18% ± 18.12 and 41.50% ± 20.23 in the T-group and C-group, respectively. The finding of this study illustrates the antimicrobial properties of totarol and its product against prevalent wound pathogens. These results suggest the potential of totarol as an adjunctive option for canine wound care.

## 1. Introduction

Open wounds are injuries that disrupt the continuity of skin and expose internal tissues. Common types of open wounds in dogs include traumatic wounds from vehicles, bite wounds, lacerations from sharp objects, gun shots, and incised wounds [1]. A clean surgical wound is expected to heal within a predicable timeframe with primary closure. However, most open wounds encountered in small animal practices often contain foreign material and devitalized tissue, increasing the risk of infection and leading to delayed wound healing known as chronic wounds. Proper wound management, including appropriate cleansing, the debridement of necrotic tissue, and regular dressing changes, can reduce bacterial colonization and later promote faster healing [2]. Additionally, a review article suggested that standard wound management involving topical antibiotics or essential oils can enhance the healing process by removing necrotic tissue and reducing bacterial load [3].

The prevalence of antibiotic-resistant bacteria is a significant threat to both health care and veterinary practice. The extensive and inappropriate use of antimicrobials has led to the rise of multidrug-resistant bacterial strains, posing treatment problems in clinical practice. Several studies have shown that the active ingredients in plant extracts possess antimicrobial activity and promote wound healing through various mechanisms. For instance, *Moringa stenopetala*, *Azadirachta indica*, and *Psidium guajava* extracts have demonstrated antimicrobial and antibiofilm activity against methicillin-resistant *Staphylococcus aureus* (MRSA) [4,5]. Lavender oil extract has been found to enhance the expression of TGF-β in wound lesions, benefiting collagen synthesis, fibroblast differentiations, and wound contraction [6]. *Clematis simensis* extract has been associated with increased epithelization and wound contraction [7]. Totarol is a natural extract derived from the heartwood of the tōtara tree (*Podocarpus totara*) in New Zealand [8], as well as other *Podocarpus* species [9] and *Juniperus* species [10]. It has a long history of use in Māori medicine for treating fevers, sores, piles, constipation, gonorrhea, syphilis, and lesions [11]. Traditionally, totarol has been used in veterinary medicine to treat canine distemper and gall sickness in cattle [12,13]. Totarol is a hydrocarbon with a single phenolic moiety and an isopropyl group ortho to the hydroxyl group, chemically identified as 14-isopropyl-8,11,13-podocarpatrien-13-ol, which is an extremely potent inhibitor of bacteria [14,15]. Totarol exhibits broad-spectrum antibacterial properties, likely mediated through interactions with bacterial membranes [14], leading to the inhibition of bacterial proliferations [16]. Additional studies have reported that totarol inhibits oxygen consumption in the respiratory chain [17], disrupts the cytoplasmic membrane [18], and reduces multidrug efflux pump activity [19]. In vitro studies demonstrate that the phytochemicals of totarol possess potent antibacterial activity against *Staphylococcus aureus*, *Streptococcus mutans*, and *Propionibacterium acne* [15]. Furthermore, totarol has been shown to inhibit the growth of *Streptococcus gordonii* and mixed oral bacteria [20]. It also inhibits the secretion of the exotoxins ∝-hemolysin, staphylococcal enterotoxin A and staphylococcal enterotoxin B [21]. Moreover, in a clinical case study involving human subjects, the medicinal use of totarol-containing moisturizer resulted in significant improvements in both the size and extent of lesions, as well as a reduction in inflammation associated with acne vulgaris [22]. However, the use of totarol as an alternative treatment for open wounds in veterinary medicine remains limited.

This study aims to assess the in vitro susceptibility of common bacteria found in clinical open wounds to totarol. Additionally, it aims to clinically evaluate the wound healing effects of standard wound care together with the use of commercial totarol products compared with standard wound care alone in client-owned dogs with open wounds.

## 2. Materials and Methods

Study design: This randomized controlled clinical trial evaluates the efficacy of standard wound care alone versus totarol adjunct to standard wound care for healing open wounds in dogs. The study was approved by the Institutional Animal Care and Use Committee of Khon Kaen University (IACUC-KKU 50/66). Owners were informed about the study protocol, consented to their dogs’ participation, received subsidized treatment costs, and signed the consent forms before the study commenced. Selected cases were allocated to either in-patient or out-patient wound management. Dogs were randomly assigned to the control group (C-group) or the treatment group (T-group) using Microsoft Excel. The owners and the data analyst were blinded for the group allocations.

Samples: The study was conducted between May 2023 and February 2024 at the Veterinary Teaching Hospital, Khon Kaen University (VTH-KKU). The VTH-KKU offers services for small animals, exotic animals, and large animals while also serving as a teaching facility for veterinary students in clinical practice. The VTH-KKU employs over 40 veterinarians and 60 support staff, managing approximately 120–150 outpatient cases per day. The cases for inclusion in the study were selected from the client-owned dogs presenting at the VTH-KKU with open wounds of at least 1 cm^2^, resulting from various causes, including vehicle trauma, bite wounds, and surgical wound dehiscence. These dogs underwent general physical examinations, complete blood counts, and serum biochemistry analyses to ensure their overall health. The dogs were amenable for wound dressing without sedation or general anesthesia (a pain score less than 2/4 using Colorado State University canine acute pain scores) [23], and those with underlying medical conditions, including severe anemia, malnutrition, diabetes mellitus, and other disorders known to impact wound healing, were excluded from the study. Of the twenty-three dogs initially considered, four were subsequently excluded from the study: one dog underwent surgical wound closure before the end of the study; another showed an increase in the wound size due to self-licking; and two dogs were lost to follow-up, discontinuing their wound dressings before the study’s completion. Therefore, nineteen dogs were included in the final analysis. The sample comprised eight males and eleven females, with an average age of 5.67 ± 3.66 years (ranging from 1 to 10 years), and an average wound age of 1.66 ± 1.16 weeks (ranging from 1 to 5 weeks). The wounds were attributed to various causes, including bite wounds (26.32%), vehicle accidents (36.84%), surgical wound dehiscence (15.79%), and other causes (21.05%).

Bacterial culture and minimum inhibitory concentration (MIC) determination: The MICs determination protocol involved collecting wound samples from clinical cases at the VTH-KKU between May 2023 and February 2024. Upon collection, wound samples underwent bacterial isolation, using blood agar and MacConkey agar plates. Bacterial strains were then identified utilizing the Vitek2 system (Table 1). Subsequently, a disk diffusion susceptibility test was performed. Ten isolated strains were selected for further analysis, including eight strains commonly found in dog wounds, along with reference strains of *Staphylococcus aureus* ATCC 25923 and *Escherichia coli* ATCC 25922. MICs were determined using a slightly modified broth microdilution method [24]. To prepare a stock solution of totarol in absolute ethyl alcohol, 0.208 g of totarol powder (Alaron Products Ltd., Nelson, New Zealand) was initially dissolved in 200 µL of absolute ethyl alcohol to make it a completely clear yellow solution. This solution was then mixed with cation-adjusted Mueller Hinton broth (CAMHB) (Cat. No. 212322, BD, Becton, Dickinson and Company, Pont-de-Claix, France). The resulting stock solutions underwent 2-fold serial dilutions in CAMHB to obtain active ingredient concentrations ranging from 2048 to 0.25 µg/mL. The commercial totarol product (VetZ PetZ™ Antinol^®^ Skin, Thaiva Laboratories, Bangkok, Thailand), which contains 0.3% (*w*/*v*) totarol as an active ingredient, is prescribed for animal skin care. It offers anti-infection and anti-inflammatory benefits and soothes hot spot. A 2-fold serial dilution was also performed to match the totarol concentrations in the tested solution.

A 0.5 McFarland bacterial suspension was prepared and adjusted to a final concentration of 5 × 10^5^ CFU/mL. Each test was conducted in triplicate for each pathogen by inoculating 10 µL of the adjusted bacterial suspension into 90 µL of the respective test solutions. The microplates were incubated at 35 ± 2 °C for 20 h. The MIC endpoints were assessed using a spectrophotometer. Absorbance measurements were recorded at 600 nm using a BioTek Epoch 2 spectrophotometer, with a 0.05 absorbance cut-off point (BioTek Instruments, Inc., Winooski, VT, USA).

To verify the MIC methods and evaluate the antimicrobial properties of the absolute ethyl alcohol, three biological replicates were conducted using two standard pathogens: *Staphylococcus aureus* ATCC 25923 and *Escherichia coli* ATCC 25922.

Treatment: All dogs received a standardized medical treatment, which included an intravenous administration of empirical antibiotics such as cefazolin 20–25 mg/kg q8h (CEFABEN^®^, L.B.S. Laboratory Ltd., Bangkok, Thailand) or amoxicillin–clavulanate 15–25 mg/kg, q8h (AMK^®^, North China Pharmaceutical Co., Ltd., Shijiazhuang, China). Oral antibiotics, such as cephalexin 20–25 mg/kg, q12h or amoxicillin–clavulanate 15–25 mg/kg, q12h, were also administered until sensitivity results were obtained, and later, an appropriate antibiotic was prescribed based on the sensitivity results. In the C-group, standard wound care involved lavage with normal saline, using 50 mL for each cm^2^ of the wounds with an 18-gauge needle attached to a 20 mL syringe and a three-way stopcock attached to a sterilized intravenous fluid line. The wound was then covered with a wound dressing or a bandage once daily for 1 week [25,26]. In the T-group, standard wound care was supplemented with the application of a totarol product at a dosage of 0.3 mL (2 sprays)/25 cm^2^ of the wound before wound dressing or bandaging once daily for 1 week. In addition, each dog wore an Elizabethan collar to prevent wound licking until the end of the study or until the wound had healed.

Wound healing measurements: The wound healing assessment utilized a combination of digital imaging and image analysis, widely regarded as the most accurate and reliable methods for wound area measurement in clinical practice [27]. The wounds were photographed using a 12-megapixel digital camera equipped with a calibrated scale inside the photo frame. Photographs were taken at a distance of approximately 15 cm from the wound and at the same angle for consistency in wound healing evaluation on days 1–7 by the same operator. The wound area was calculated using ImagJ software version 1.54 (National Institutes of Health [NIH image]) [28,29]. The initial wound area was measured at baseline, and the change in wound area compared to day 1 was reported as the percentage of wound area reduction. This percentage was calculated using the previously published formula [30,31,32] to ensure consistency and accuracy in the assessment process:$$\%\mathrm{Wound}\;\mathrm{area}\;\mathrm{reduction}=\frac{\left(W_{1}\right)-(W_{x})}{\left(W_{1}\right)}\times 100\%$$

W_1_ = initial wound area;

Wx = area on measurement day.

Data analysis: The MIC values were reported as a description of the totarol assessed against standard pathogens and clinical wound pathogens. General information of the subjects, including patient age, wound age, and initial wound area, was compared between the C-group and the T-group by independent T-tests. An evaluation of the average wound area reduction in the clinical cases, including both the C-group and the T-group, was statistically analyzed using the repeated linear mixed model. The full model included fixed effects, such as the treatment protocol, and the day in the healing evaluation, along with their interaction effect. A random intercept was included to accommodate within-dog variability, and a random slope on the repeated day for the healing evaluation was assigned with an unstructured covariance structure. The simple effect comparison between groups at each time point was tested by post-estimation analysis (CONTRAST command). Significance was determined at a *p*-value of <0.05.

## 3. Results

The antimicrobial effects of absolute ethyl alcohol greater than 60-fold did not exhibit antimicrobial properties in the MIC results. The MIC results of two standard pathogens were consistent across three replicates, as shown in Table 2.

### 3.1. Antimicrobial Activity of Totarol

The in vitro study revealed that totarol exhibited an antimicrobial activity against clinical wound pathogens (Table 3) in dogs. The MICs of totarol dissolved in absolute ethyl alcohol were found to be 4 µg/mL, indicating the potent inhibition of Gram-positive wound pathogens. However, Gram-negative pathogens seemed to require a higher concentration of totarol for inhibition. A comparison between totarol dissolved in absolute ethyl alcohol and commercial totarol product suggests differences in susceptibility.

### 3.2. Clinical Assessment of the Wound Healing

There was no significant difference in wound age between the T-group and C-group (*p* = 0.57), with an average wound age of 1.80 ± 1.30 and 1.50 ± 1.03 weeks, respectively. Although there was a trend toward differences in the initial wound area between the two groups (Figure 1), this difference was not statistically significant (*p* = 0.07). The average initial wound area size was 7.19 ± 4.21 cm^2^ for the T-group and 12.75 ± 7.97 cm^2^ for the C-group. To account for this variability, the wound area was adjusted to equalize the wound size during the data analysis. After day 2 to day 7 of treatment, there was a significant difference (*p* < 0.05) in the percentage of wound area reduction between the T-group and C-group, as indicated in Table 4. Additionally, both groups exhibited a significant reduction in the percentage of wound area over time (*p* = 0.00), with the coefficients of the percentage of wound reduction change being 8.93 and 6.76 for the T-group and C-group, respectively. The wounds of five dogs were sutured after 7 days because the wounds were suitable to carry out this procedure. For the remaining wounds (fifteen dogs), the average time to wound closure was approximately of 17.73 days (with a range of 9–34 days), and there was no significant difference between the two groups at the end.

## 4. Discussion

Our study demonstrates the positive antimicrobial activity of totarol against wound pathogens. This finding is supported by existing articles that highlight the antibacterial properties of totarol in medical products [20,33]. Additionally, totarol has also been effective against *Staphylococcus aureus* as a natural preservative in the food industry [18]. Regarding the MICs’ determination, totarol dissolved in absolute ethyl alcohol showed greater inhibition against Gram-positive pathogens with MIC values of 4 µg/mL, as shown in Table 2 and Table 3. These results are consistent with recent findings regarding its efficacy against Gram-positive bacteria, including wild-type *Staphylococcus pseudintermedius* and *Staphylococcus coagulans* isolates [8]. Furthermore, various studies have found totarol to be effective against Gram-positive pathogens, such as *Staphylococcus aureus*, *Bacillus subtilis*, *Brevibacterium ammoniagenes*, *Streptococcus mutans*, and *Propionibacterium acne,* with the MIC values of 0.39–1.56 µg/mL [15]. However, a high concentration of totarol is required to inhibit Gram-negative bacteria, potentially due to the robust defensive systems of their outer membranes, which contribute to antimicrobial resistance [34,35]. This study observed that the MIC values of the commercial totarol product ranged between 512 and 1024 µg/mL, which significantly differed from those of the totarol solution in absolute ethyl alcohol. However, the authors cannot provide a clear explanation for this difference. This disparity may be attributed to variations in solvent composition, which can influence test outcomes, as well as the varying susceptibility of isolated pathogens to these solvents.

The C-group comprised the control group/the standard wound-care-alone group, while the T-group comprised totarol supplementing the standard wound care. 

Our clinical study data demonstrate the effectiveness of totarol as an adjunct to standard wound care in enhancing the wound healing process. Our findings indicated a significant increase in the rate of wound area reduction when totarol was added to standard wound care compared to standard wound care alone, starting from day 2 of treatment (Table 4). These results are consistent with our in vitro study, which emphasized the promising antimicrobial potential of totarol and its role in improving wound healing. To our knowledge, only one previous study has assessed the clinical use of a totarol-containing spray or clay to treat canine superficial pyoderma caused by methicillin-susceptible *Staphylococcus pseudintermedius*. In this trial, the dogs receiving totarol demonstrated positive effects on clinical signs, including erythema, exudation, pruritus, hair loss, and the depth of lesions [36]. Therefore, our study represents the first clinical trial of a veterinary commercial totarol product for open wound management in dogs. Additionally, clinical case studies in humans have indicated improvement in the healing of skin acne lesions and a reduction in inflammation through the application of totarol-containing moisturizers [22]. A recent study has also demonstrated that tissue adhesives containing totarol could eliminate the presence of aerobic bacteria, yeast, and mold counts [33]. In our clinical trial, the use of totarol combined with standard wound care enhanced faster wound healing in dogs.

The small sample size in this study may have contributed to the unequalness of the initial wound sizes. The variable wound model of the clinical cases, which encompasses factors such as wound cause, wound age, and wound shape, may have influenced the wound healing process. Although efforts were made to adjust the initial wound area sizes to equalize wound areas before comparison in the data analysis, the inherent variability in the clinical cases could have impacted the study’s outcomes. Further studies could focus on exploring the clinical use of totarol in treating small animal skin diseases, including superficial dermatitis, deep pyoderma, or otitis externa. Investigating the efficacy and safety of totarol in these conditions would provide a valuable insight into its potential applications in veterinary practice and contribute to the existing body of knowledge on alternative therapeutic approaches.

## 5. Conclusions

The current study demonstrated that totarol exhibited significant antimicrobial activity against clinical wound pathogens in dogs. Additionally, our clinical trial provides evidence that the topical application of totarol enhances wound healing by accelerating the rate of wound area reduction. These findings suggest that totarol could serve as a promising adjunct to standard wound care for the treatment of open wounds in dogs.

## Figures and Tables

**Figure 1 vetsci-11-00437-f001:**
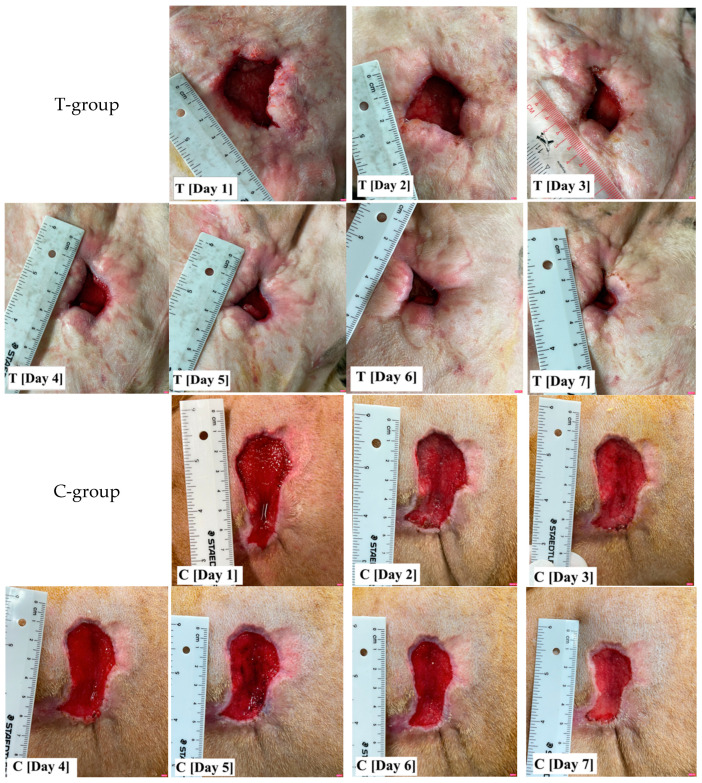
The wound photographs show the variability of the initial wound size and shape. Following wound adjustment, the percentage of wound area reduction in the T-group, where standard wound care was supplemented with totarol, was found to be significantly greater than that of the C-group, which involved standard wound care alone, after day 2 of treatment.

**Table 1 vetsci-11-00437-t001:** A summary of selected cases and their wound pathogens strains.

Subjects	Gender	Age (Years)	Wound Cause	Wound Age (Weeks)	Initial Wound Area (cm^2^)	Group of Treatment	Wound Pathogens
Dog 1	Female	10	vehicle accidents	1	9.70	Standard wound care	*Escherichia coli*
Dog 2	Female	10	surgical wound dehiscence	1	2.77	Standard wound care + totarol	*Escherichia coli*
Dog 3	Female	2	surgical wound dehiscence	2	11.26	Standard wound care + totarol	*Proteus mirabilis* *Acinetobacter baumannii*
Dog 4	Female	10	other causes	1	7.21	Standard wound care	*Proteus mirabilis*
Dog 5	Male	7	bite wounds	2	29.01	Standard wound care	*Pasteurella canis*
Dog 6	Male	10	bite wounds	3	11.19	Standard wound care + totarol	*Staphylococcus pseudintermedius*
Dog 7	Male	5	other causes	1	8.19	Standard wound care + totarol	*Proteus mirabilis* *Escherichia coli*
Dog 8	Male	2	vehicle accidents	1	-	Excluded	*Staphylococcus haemolyticus*
Dog 9	Female	3	vehicle accidents	2	9.77	Standard wound care	*Pseudonomas aeruginosa*
Dog 10	Female	9	bite wounds	1	-	Excluded	*Staphylococcus schleiferi*
Dog 11	Male	4	other causes	5	6.63	Standard wound care + totarol	*Enterococcus faecium*
Dog 12	Female	10	other causes	1	5.31	Standard wound care + totarol	*Proteus mirabilis*
Dog 13	Male	2	vehicle accidents	2	3.55	Standard wound care	*Proteus mirabilis*
Dog 14	Female	9	bite wounds	1	1.51	Standard wound care + totarol	*Klebsiella pneumoniae*
Dog 15	Female	6	bite wounds	1	2.00	Standard wound care + totarol	*Sphingomonas paucimobilis*
Dog 16	Male	4	surgical wound dehiscence	2	9.79	Standard wound care + totarol	*Escherichia coli*
Dog 17	Male	9	bite wounds	1	-	Excluded	*Proteus mirabilis*
Dog 18	Male	2	vehicle accidents	2	13.24	Standard wound care + totarol	*Proteus mirabilis*
Dog 19	Female	1	vehicle accidents	1	20.71	Standard wound care	Coagulase-positive staphylococcus
Dog 20	Female	1	vehicle accidents	1	16.32	Standard wound care	*Staphylococcus epidermidis*
Dog 21	Male	3	vehicle accidents	1	11.03	Standard wound care	*Staphylococcus aureus*
Dog 22	Female	6	bite wounds	1	-	Excluded	*Enterobacter cloacae*
Dog 23	Female	10	bite wounds	2	7.38	Standard wound care	*Pseudonomas aeruginosa*

**Table 2 vetsci-11-00437-t002:** The MIC values of absolute ethyl alcohol control, totarol dissolved in absolute ethyl alcohol, and commercial totarol product were assessed against two standard pathogens with three replicates each.

Standard Pathogens	Replications	MIC Endpoints (µg/mL)		
		Absolute Ethyl Alcohol Control	Totarol Dissolved in Absolute Ethyl Alcohol	Commercial Totarol Product
*S. aureus* ATCC 25923	1	1024	4	512
	2	1024	4	512
	3	1024	4	512
*E. coli* ATCC 25922	1	>2048	>2048	512
	2	>2048	>2048	512
	3	>2048	>2048	512

**Table 3 vetsci-11-00437-t003:** The MIC values of totarol dissolved in absolute ethyl alcohol and a commercial totarol product were assessed against clinical wound pathogens.

Clinical Wound Pathogens	MICs of Totarol (µg/mL)	
	Totarol Dissolved in Absolute Ethyl Alcohol	Commercial Totarol Product
*Staphylococcus epidermidis* (*n* = 1)	4	512
*Staphylococcus pseudintermedius* (*n* = 1)	4	512
*Staphylococcus haemolyticus* (*n* = 1)	4	1024
*Staphylococcus aureus* (*n* = 1)	4	512
Coagulase-positive staphylococcus (*n* = 1)	4	512
*Enterobacter cloacae* (*n* = 1)	256	512
*Proteus mirabilis* (*n* = 1)	512	1024
*Klebsiella pneumoniae* (*n* = 1)	512	512

**Table 4 vetsci-11-00437-t004:** The averages and standard deviations of the percentage of wound area reduction were compared between the T-group (standard wound care with totarol) and the C-group (standard wound care alone) for each treatment time period. The T-group showed a significantly greater wound area reduction than the C-group after day 2 of treatment (*p* < 0.05).

Group	Day 1	Day 2	Day 3	Day 4	Day 5	Day 6	Day 7	Average
T-group	0.00	25.10 ± 16.16	33.60 ± 17.65	45.48 ± 23.02	52.96 ± 22.67	58.53 ± 21.71	69.18 ± 18.12	40.69 ± 28.04
C-group	0.00	8.53 ± 8.40	11.85 ± 9.22	21.43 ± 17.22	27.64 ± 17.45	37.24 ± 18.69	41.50 ± 20.23	21.17 ± 20.23
*p-value* *	-	0.017	0.003	0.002	0.003	0.020	0.006	

* The simple effect comparison between the groups at each time point was tested by post-estimation (CONTRAST command) analysis. The significance was determined at a *p*-value of <0.05.

## Data Availability

The original contributions presented in the study are included in the article, further inquiries can be directed to the corresponding author.

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
