# Peer review of "Evaluation of Totarol for Promoting Open Wound Healing in Dogs"

_vetsci, 2024, doi:10.3390/vetsci11090437_

Round 1

Reviewer 1 Report

Comments and Suggestions for Authors

I would like to thank the journal for the opportunity of reviewing this manuscript, which is really interesting and of great practical application. I would also clarify some aspects of this manuscript with the authors.

Lines 30-32: You should differentiate whether MIC values are related to totarol diluted in ethyl alcohol or to commercial totarol.

Introduction: The authors should include more information on totarol (at least of traditional use). Explanations about its pharmacological properties are really short.

Materials & Methods: Sample: the authors should provide information on the characteristics of the Veterinary Teaching Hospital (animals received, practitioners….) to assess if the sample size is coherent. For how long did they work to get that sample? You should also reference the pain scale used (scores 2-4).

You should provide with more information on the commercial product. I’m not be able to find it. Moreover, quantitative and qualitative composition should be described (active ingredients).

You should also confirm whether concentrations of totarol in ethyl alcohol and commercial totarol were the same. There is a lack of information about it.

Since a microdilution procedure was followed, information on a negative control with only ethyl alcohol should be provided. Why did you decide to dilute in ethyl alcohol, as other authors (referenced in the manuscript) used DMSO. Ethyl alcohol is well known for their antibacterial properties, with established MIC against various microorganisms. How can you establish if the antibacterial activity is partially due to ethyl alcohol.

Did microorganisms finally disappear from the wounds? MICs of 256-512 µg/mL should be considered resistant rather than sensitive in terms of susceptibility. Why did the authors attribute the effect to totarol? Would it be possible to attribute antibacterial activity to other active ingredients present in the commercial formulation (synergistic action)?

Results. Lines 192-193. You give the initial area of the wound, but not the final one.

Lines 196-198: What means this coefficient?

It’s not appropriate to express significant differences as “statistically significant”, and when a p-value is given, the corresponding performed test should also be provided.

Could you please give some information on the final outcome of the treatment and not only partial results after 7 days?

Limitations: sample size is the main limiting factor, as it’s not possible to correlate the results obtained with any other variable.

Lines 260-263: Not sure if this sentence is appropriate, as they have administered different antibiotics to reinforce wound healing (lines 126-134).

The authors should take care of the overall formatting of the manuscript, as some paragraphs have a different font size.

Author Response

Response to reviewer-1 and comments

1. Summary The authors would like to thank you for your thoughtful comments. We have carefully revised the manuscript following your comments and questions. Below are specific point-by-point responses about the questions raised. We have highlighted the corrections and additions in the manuscript in red text. 

Reviewer 2 Report

Comments and Suggestions for Authors

The authors use totarol as a treatment for various wounds in domestic dogs and demonstrate improved wound closure in dogs that received daily application of a totarol wound spray.

Major Comments:

1. The background should discuss the chemical structure and composition of totarol

2. The MIC assay needs to be explained clearly in the methods section.

3. Table 2. MIC results. What was the number of biological replicates (n=?)? What is the SD? There needs to be an ethyl alcohol vehicle control so that you know what is the specific effect of totarol. Ethyl alcohol on its own is anti-bacterial. What is the vehicle in the commercial product? 

4. Table 3. Wound area reduction. Are there any time points later than 7 days?  Were there differences in time to full wound closure?

5. Discussion. The authors should comment on individual wounds. For example, was there a difference in healing between wounds that were infected with different bacteria (Proteus vs Staph vs Pseudo, etc)?

6. Discussion. The authors need to discuss why their MICs of totarol are so much higher than published reports (https://enviromicro-journals.onlinelibrary.wiley.com/doi/10.1111/j.1365-2672.1996.tb03233.x)

Minor comment:

1. line 157 states "The MIC values were reported by descriptive of totarol dilution." What does this mean? Please clarify.

Comments on the Quality of English Language

Needs to be edited for English grammar 

Author Response

Response to reviewer-2 and comments
The authors would like to thank you for your thoughtful comments. We have carefully revised the manuscript following your comments and questions. Below are specific point-by-point responses about the questions raised. We have highlighted the corrections and additions in the manuscript in red text.  
"Please see the attachment."

Round 2

Reviewer 1 Report

Comments and Suggestions for Authors
  1. I would recommend to include the characteristics of the veterinary hospital (They say that it offers services for small animals, exotic animals, and large animals, while also serving as a teaching facility for veterinary students in clinical practice. The VTH-KKU employs over 40 veterinarians and 60 support staff, managing approximately 120-150 outpatient cases per day) as it gives an idea of the actual working load of the hospital. 
  2. Finally, I would describe the commercial product used to  treat animals (VetZ PetZ) in the section where treatments applied to animals are described. Moreover, in this case 0.3 % should be followed by the abbreviation (w/v).

Author Response

The authors would like to thank you gratitude to the reviewers for taking the time to reviewing of our manuscript. The suggestions and comments provided are invaluable, and we have made careful revisions to improve the quality of our work. Please see the attachment.

Reviewer 2 Report

Comments and Suggestions for Authors

Response to comment 1 is fine

Response to comment 2: Add the length of incubation time with totarol and the temperature of the incubation. Add how the bacterial inhibition was measured. Did you measure absorbance with a spectrophotometer? If so, explain clearly

Response to comment 3:

Add "n = 1" to the Table

A 1:100 dilution of absolute ethyl alcohol could still have an effect. Please indicate that you did not include a 1:100 absolute ethyl alcohol vehicle control and discuss this in the discussion

Response to Comment 4 is fine

Response to Comment 5 is fine

Response to comment 6 is fine

Response to comment 7 It is still not clear what "reported by descriptive of totarol dilution" means. Does this mean they just looked at the tubes and made a decision? Did they use a spectrophotometer to quantitatively measure bacterial density? The method needs to be clearly explained. 

After review of the revisions, the lack of biological replicates, the lack of a vehicle control, and no clear explanation for how MIC values were determined  have reduced this reviewer's overall recommendation.
=======================

 Explain the MIC assay. How was it quantified? Did they use a spectrophotometer Redo the MIC assay with at least n=3 biological replicates

Include a vehicle (ethyl alcohol) control in the MIC assay Comments on the Quality of English Language

minor English grammar mistakes

Author Response

(The authors gave the same response as above.)
